# A Novel Methodology for GB-SAR Estimating Parameters of the Atmospheric Phase Correction Model Based on Maximum Likelihood Estimation and the Gauss-Newton Algorithm

**DOI:** 10.3390/s24175699

**Published:** 2024-09-01

**Authors:** Xiheng Li, Yu Liu

**Affiliations:** 1School of Electronic and Information Engineering, Chongqing Three Gorges University, Chongqing 404130, China; remoteradar@stumail.sanxiau.edu.cn; 2Key Laboratory of Geological Environment Monitoring and Disaster Early Warning in Three Gorges Reservoir Area, Chongqing 404120, China

**Keywords:** ground-based synthetic aperture radar (GB-SAR), atmospheric phase correction (APC), maximum likelihood estimation (MLE), Gauss-Newton algorithm, Matthews and Davies algorithm

## Abstract

Atmospheric phase error is the main factor affecting the accuracy of ground-based synthetic aperture radar (GB-SAR). The atmospheric phase screen (APS) may be very complicated, so the atmospheric phase correction (APC) model is very important; in particular, the parameters to be estimated in the model are the key to improving the accuracy of APC. However, the conventional APC method first performs phase unwrapping and then removes the APS based on the least-squares method (LSM), and the general phase unwrapping method is prone to introducing unwrapping error. In particular, the LSM is difficult to apply directly due to the phase wrapping of permanent scatterers (PSs). Therefore, a novel methodology for estimating parameters of the APC model based on the maximum likelihood estimation (MLE) and the Gauss-Newton algorithm is proposed in this paper, which first introduces the MLE method to provide a suitable objective function for the parameter estimation of nonlinear far-end and near-end correction models. Then, based on the Gauss-Newton algorithm, the parameters of the objective function are iteratively estimated with suitable initial values, and the Matthews and Davies algorithm is used to optimize the Gauss-Newton algorithm to improve the accuracy of parameter estimation. Finally, the parameter estimation performance is evaluated based on Monte Carlo simulation experiments. The method proposed in this paper experimentally verifies the feasibility and superiority, which avoids phase unwrapping processing unlike the conventional method.

## 1. Introduction

Ground-based synthetic aperture radar (GB-SAR) is an active microwave detection technique that originates from interferometric synthetic aperture radar (InSAR). GB-SAR is characterized by its small scale, high flexibility, and high spatial and temporal resolution. Thus, it has been widely used in monitoring landslides [1,2,3,4,5,6], glacier movement [7], mine subsidence [8] and civil structures [9]. Depending on different geographical environments, an appropriate correction method should be selected to improve the accuracy of GB-SAR deformation monitoring [10], and maximum likelihood estimation (MLE) has been used in some studies [11,12]. However, the atmospheric phase screen (APS) seriously affects the monitoring accuracy. Research shows that a 1% change in relative humidity 1 km from the radar site at 20 °C will lead to a 2 mm measurement error [13]. Therefore, the effect of APS should be minimized to improve the accuracy of GB-SAR deformation measurements.

The three most basic correction methods for APS caused by changes in atmospheric conditions are ground control point correction, meteorological data correction, and parametric modeling methods. The ground reference point correction method is a correction method that utilizes the spatial interpolation of the atmospheric phase values of the reference points in the scene to obtain the atmospheric phase values at all points [14,15]. The meteorological data correction method involves establishing a meteorological station within the observation scene and using meteorological data (temperature, humidity, and atmospheric pressure) to quantitatively estimate the atmospheric phase based on the atmospheric-refractive index model [16]. Meanwhile, in a new study, researchers simulate the relevant meteorological parameters using the ensemble scheme of the stochastic physic perturbation tendency based on the weather research forecasting model [17]. The parametric modeling method is a correction method that models the atmospheric phase and then estimates the model parameters with the help of the permanent scatterer (PS) technique to obtain the atmospheric phase values at all PS points [18]. Under ideal conditions, where the atmosphere varies uniformly in space, the atmospheric phase can be modeled as a component that varies linearly with the slope distance [19]. However, this linear relationship is sometimes affected by phase ramps such as orbital errors, tidal loading, etc. There is a new approach to estimate the transfer function of vertical stratification phase delays and the transfer function of phase ramps [20]. Focusing on stratified atmospheric delay correction in the small-amplitude displacement field of small-to-moderate earthquakes, researchers developed a Simple-Stratification-Correction approach based on the empirical phase-elevation relationship and spatial properties of the troposphere, via an equal-size window segmentation [21]. 

However, in general, meteorological conditions change with time, and the atmosphere varies non-homogenously in space, resulting in complex null variability in the atmospheric phase and leading to the inability to build a suitable multiparameter model to simulate the atmospheric phase. In addition, many conventional atmospheric phase correction (APC) methods, which are based on homogeneous atmospheric phases, cannot effectively correct atmospheric phase, so more complex and comprehensive APC methods are needed. Therefore, researchers have proposed an improved correction method for the nonlinear atmospheric phase of GB-SAR images [22]. Researchers use advanced deep learning algorithms to individually model and mitigate topography-dependent atmospheric delay [23]. Partition correction is a feasible path to improve APC accuracy for complicated APS, but the over-fitting problem cannot be ignored. Therefore, a clustering partition method based on the normal vector of APS is proposed, which can partition the complicated APS more reasonably and then perform APC based on the partition results [24]. Finding an accurate and reliable interval for preparing mean deformation rate maps remains challenging; there is an application consisting of three unsupervised clustering algorithms for determining the best interval from synthetic aperture radar (SAR)-derived deformation data, which can be used to interpret long-term deformation processes [25]. Researchers have combined topography and spatial data information, proposing a topography-dependent APC method based on the Multi-Layer Perceptron neural network model [26]. Other researchers adopted the concept of a coordinate system for modeling. They proposed a 2D APC method based on a polar coordinate system, which can be used to solve non-homogenous APC in the azimuthal direction; they also proposed a 3D APC method based on a right-angled coordinate system, which can address non-homogenous APS in three directions of non-homogenous APS [27]. In the process of APC, direct least-squares fitting of the interferometric phase is not feasible, and because phase unwrapping is mostly carried out based on the phase continuity assumption, the wrapped phase contains jump points [28]. Therefore, the conventional least-squares method (LSM) is not applicable to discrete PS points in discontinuous interferometric synthetic aperture radar (D-InSAR) with wrapped phases, and poorer phase unwrapping introduces additional unwrapping error, which in turn affects the accuracy of GB-SAR. Therefore, researchers have developed a likelihood function model to bypass the phase unwrapping process and directly estimated the parameters to account for the error phase [29].

In summary, this paper proposes MLE and a Gauss-Newton algorithm to estimate parameters of the APC model and remove the APS to improve the measurement accuracy of GB-SAR deformation monitoring. First, according to the distance from the monitoring target to the radar site, the high-quality PS points are divided into proximal PS points and distal PS points by K-means clustering, and the proximal APC model and distal APC model are proposed. The proximal APC model based on the polar coordinate system and the digital elevation model (DEM) is applied to address the non-homogenous APS at the proximal distance. This model is applicable to regions with relatively flat terrain and observation scenes with small radar range slopes, and the far-end APC model based on the right-angle coordinate system addresses non-homogenous APS in three directions (one vertical direction and two horizontal directions) at long distances, mainly targeting regions with relatively steep terrain and observation scenes with large radar range slopes. Second, the objective function of the near-end and far-end correction parameter model is derived based on the MLE method; the parameters of the objective function are iteratively estimated by the Gauss-Newton algorithm with appropriate initial values. The Gauss-Newton algorithm is optimized by the Matthews and Davies algorithm to improve the accuracy of data processing, and Monte Carlo simulation experiments are used to compare the Cramer-Rao lower bound (CRLB) and mean square error (MSE) to evaluate the performance of parameter estimation. Finally, real terrain data are used to compare this paper’s method with the general APC method and the APC method based on the LSM and the Newton algorithm, which verifies the validity and superiority of this paper’s method.

The structure of this paper is as follows: Section 2 proposes two models for APC at the near and far ends, derives the likelihood functions of the two models in detail, and uses the Gauss-Newton method as the optimization algorithm to iteratively estimate the model parameters. Section 3 provides a brief introduction of the experimental site in the Chair Hill area of Chongqing and shows the algorithmic steps and overall framework of this paper. Section 4 and Section 5 present the experimental validation and conclusions of the methodology of this paper, respectively.

## 2. Materials and Methods

### 2.1. PS Selection

When deformation measurements are performed through pixel points in radar images, the phase quality of the pixel points directly affects the accuracy of the deformation measurements. Therefore, some high-quality pixel points, i.e., PSs, are usually required when GB-SAR performs differential interferometry. The amplitude divergence method is the method most commonly used for selecting PSs, and the ratio of the standard deviation to the mean value of the time series of the amplitude of a pixel point is the amplitude divergence index. A PS can be selected by setting a reasonable amplitude deviation threshold [30]. The expression for solving the amplitude deviation index (ADI) is shown below:(1)$$D_{A}=\frac{\delta_{A}}{m_{A}},$$
where $\delta_{A}$ is the standard deviation of the amplitude and $m_{A}$ is the mean of the amplitude; subscript A represents amplitude.

We will calculate the signal-to-noise ratio (*SNR*) and coherence factor of the PS points selected through ADI and classify them into three categories using a K-means algorithm. At the same time, we will register and interpolate the point cloud collected by the terrestrial laser scanner (TLS) with the PS points.

### 2.2. Far-End and Near-End Correction Model Functions

Based on the elevation information of the 3D model and the observed interferograms, the estimated PS phases have errors for the conventional GB-SAR deformation monitoring system, including the anomalous phase increments at the far and near ends of the distance, due to the random variations in the atmospheric parameters in the monitoring region. Therefore, this paper proposes a non-homogenous APC method based on the far-end and near-end models with the following expression:(2)$$\Delta \varphi_{\mathrm{corr}}=\arg\left\{\exp\left[j\int_{1}^{w}\left(\Delta \varphi_{i}-\Delta \varphi_{APC1}^{i}\right)dr\right]+\exp\left[j\int_{w}^{n}\left(\Delta \varphi_{i}-\Delta \varphi_{APC2}^{i}\right)dr\right]\right\},$$
where $\Delta \varphi_{i}$ is the interferometric phase affected by atmospheric variations and $\Delta \varphi_{APC1}$ and $\Delta \varphi_{APC2}$ are the estimation functions for the APC at the far and near ends, respectively; j represents the imaginary unit. It is worth noting that w is the boundary that divides the model region into the far end and near end by K-means clustering; the distance between the PS points and the radar center aperture is less than w, which is called the near-end region. Conversely, it is called the far-end region; n represents the farthest distance from PS points to the center aperture of the radar.
(3)$$\Delta \varphi_{APC1}^{i}=\beta_{1}r_{i}+\beta_{2}r_{i}\cdot \theta_{i}+\beta_{3}r_{i}\cdot h_{i}=\beta_{1}r_{i}+\beta_{2}l_{i}+\beta_{3}r_{i}\cdot h_{i},$$
(4)$$\Delta \varphi_{APC2}^{i}=\beta_{1}x_{i}\cdot r_{i}+\beta_{2}y_{i}\cdot r_{i}+\beta_{3}h_{i}\cdot r_{i},$$
where $r_{i}$ denotes the distance from the *i*^th^ PS to the radar center aperture, $\theta_{i}$ denotes the deflection angle of the i^th^ PS from the radar center aperture, and $l_{i}$ denotes the arc length formed between the i^th^ PS and the radar center aperture. $x_{i}$, $y_{i}$ and $h_{i}$ denote the 3D coordinates of the i^th^ PS. $\beta_{i}(i=1,2,3)$ are the parameters of the APC model to be estimated; they are crucial parameters in the methodology of this paper.

### 2.3. Gauss-Newton Algorithm for Iterative Parameter Estimation

In Equations (3) and (4) in the previous section, $\beta_{i}(i=1,2,3)$ are the three parameters of the model function to be estimated, which can be estimated by the conventional LSM. However, direct least-squares fitting of the interferometric phase is not feasible because the phase unwrapping operation is not performed. In particular, the general phase unwrapping method is prone to introducing additional unwrapping errors, which further affects the accuracy of terrain deformation measurements. To address the above problems, this paper proposes an APC processing method based on the MLE and the Gauss-Newton algorithm, taking the proximal correction model function in Equation (3) as an example. The complex exponential function is established as follows:(5)$$\exp(j\Delta \varphi_{i})=\exp\left\{j\cdot \left[\beta_{1}r_{i}+\beta_{2}l_{i}+\beta_{3}r_{i}\cdot h_{i}\right]\right\}+{\tilde{w}}_{i},$$
where $\exp(j\Delta \varphi_{i})$ denotes the complex index of the *i*^th^ PS phase. According to Equation (5), the estimated value of the *i*^th^ PS can be expressed as
(6)$$x_{i}=\exp\left\{j\cdot \left[\beta_{1}r_{i}+\beta_{2}l_{i}+\beta_{3}r_{i}\cdot h_{i}\right]\right\}+{\tilde{w}}_{i}=x_{1,i}+j\cdot x_{2,i},$$
$$x_{1,i}=\cos(\beta_{1}r_{i}+\beta_{2}l_{i}+\beta_{3}r_{i}\cdot h_{i})+{\tilde{w}}_{1,i},$$
$$j\cdot x_{2,i}=j\cdot \sin(\beta_{1}r_{i}+\beta_{2}l_{i}+\beta_{3}r_{i}\cdot h_{i})+j\cdot {\tilde{w}}_{2,i},$$
$${\tilde{w}}_{1,i}\sim N(0,\sigma^{2}),$$
$$j\cdot {\tilde{w}}_{2,i}\sim N(0,\sigma^{2}).$$

$x_{1,i}$ and $x_{2,i}$ correspond to the real and imaginary parts, respectively. The real and imaginary parts of noise are independent and Gaussian distributed random variables. Gaussian noise can be represented as a complex number, where the real part represents the actual noise and the imaginary part represents the phase of the noise. Therefore, ${\tilde{w}}_{1,i}$ and ${\tilde{w}}_{2,i}$ correspond to the Gaussian noise in the real and imaginary parts, respectively. ${\tilde{w}}_{1,i}$ and ${\tilde{w}}_{2,i}$ follow a Gaussian distribution with a zero mean and a variance of $\sigma^{2}$, where the value range of $\sigma^{2}$ is [0.15, 5].

Therefore, the likelihood function of the parameter to be estimated is as follows:(7)$$L(x;\vec{\beta })=\frac{1}{\sqrt{{\left(2\pi \sigma^{2}\right)}^{N}}}\cdot \exp\left\{-\frac{1}{2\sigma^{2}}\sum_{i=1}^{N}{\left|x_{i}-\exp(j\cdot (\beta_{1}r_{i}+\beta_{2}l_{i}+\beta_{3}r_{i}\cdot h_{i}))\right|}^{2}\right\},$$
where $\left|\cdot \right|$ denotes the absolute value operator. The following equation is minimized in order to obtain the MLE.
(8)$$\hat{\beta }=\underset{\vec{\beta }}{\arg}\min\sum_{i=1}^{N}{\left|x_{i}-\exp(j\cdot (\beta_{1}r_{i}+\beta_{2}l_{i}+\beta_{3}r_{i}\cdot h_{i}))\right|}^{2}.$$

By expanding the above equation, an expression in vector form is obtained:(9)$$f(\vec{\beta })={\left(\vec{x}-\vec{s}\right)}^{H}(\vec{x}-\vec{s})={\vec{x}}^{H}\vec{x}+{\vec{s}}^{H}\vec{s}-{\vec{s}}^{H}\vec{x}-{\vec{x}}^{H}\vec{s},$$
$$\vec{x}=\left[\begin{array}{c} x_{1}\\ x_{2}\\ \vdots \\ x_{i} \end{array}\right]=\left[\begin{array}{c} \exp(j\cdot \Delta \varphi_{1})\\ \exp(j\cdot \Delta \varphi_{2})\\ \vdots \\ \exp(j\cdot \Delta \varphi_{i}) \end{array}\right]\;\vec{s}=\left[\begin{array}{c} s_{1}\\ s_{2}\\ \vdots \\ s_{i} \end{array}\right]=\left[\begin{array}{c} \exp(j\cdot (\beta_{1}r_{1}+\beta_{2}l_{1}+\beta_{3}r_{i}\cdot h_{i})\\ \exp(j\cdot (\beta_{1}r_{2}+\beta_{2}l_{2}+\beta_{3}r_{i}\cdot h_{i})\\ \vdots \\ \exp(j\cdot (\beta_{1}r_{i}+\beta_{2}l_{i}+\beta_{3}r_{i}\cdot h_{i}) \end{array}\right].$$

H in Equation (9) denotes the conjugate transpose operator, and since ${\vec{x}}^{H}\vec{x}+{\vec{s}}^{H}\vec{s}$ is a constant, the MLE solution is equivalent to maximizing ${\vec{s}}^{H}\vec{x}+{\vec{x}}^{H}\vec{s}$ to obtain $\vec{\beta }$ as follows: (10)$$\hat{\beta }=\arg_{\vec{\beta }}\max\sum_{i=1}^{N}\left({\vec{s}}^{H}\vec{x}+{\vec{x}}^{H}\vec{s}\right).$$

Thus, the objective function of the parameter to be estimated can be obtained: (11)$$f(\vec{\beta })={\vec{s}}^{H}\vec{x}+{\vec{x}}^{H}\vec{s}=2\cos(\beta_{1}r_{i}+\beta_{2}l_{i}+\beta_{3}r_{i}\cdot h_{i}-\Delta \varphi_{i}).$$

Similarly, the objective function of the parameters to be estimated for the distal correction model is
(12)$$f(\vec{\beta })={\vec{s}}^{H}\vec{x}+{\vec{x}}^{H}\vec{s}=2\cos(\beta_{1}r_{i}\cdot x_{i}+\beta_{2}r_{i}\cdot y_{i}+\beta_{3}r_{i}\cdot h_{i}-\Delta \varphi_{i}).$$

The advantage of MLE is that the unknown parameters of its first-order derivatives are separable. However, Equation (11) shows that the objective function is modeled as a nonlinear function, which cannot be independent of parameter $\vec{\beta }$. Therefore, the likelihood function cannot be solved directly, and this paper proposes a parameter estimation method based on Gauss-Newton iteration. The objective function in the Gauss-Newton algorithm takes the form of a vector:(13)$$f={\left[f_{1}(\vec{\beta }),f_{2}(\vec{\beta }),\cdots ,f_{n}(\vec{\beta })\right]}^{T},$$
(14)$$F=\sum_{p=1}^{n}f_{p}{\left(\vec{\beta }\right)}^{2}=f^{T}f.$$

For parameters $\beta_{1},\beta_{2,},\beta_{3}$, the gradient matrix of each of its functions $f_{p}(\vec{\beta }),p=1,2,\cdots ,n$ is called a Jacobi matrix:(15)$$J=\left[\begin{array}{ccc} \frac{\partial f_{1}}{\partial \beta_{1}} & \frac{\partial f_{1}}{\partial \beta_{2}} & \frac{\partial f_{1}}{\partial \beta_{3}}\\ \frac{\partial f_{2}}{\partial \beta_{1}} & \frac{\partial f_{2}}{\partial \beta_{2}} & \frac{\partial f_{2}}{\partial \beta_{3}}\\ \vdots & \vdots & \vdots \\ \frac{\partial f_{n}}{\partial \beta_{1}} & \frac{\partial f_{n}}{\partial \beta_{2}} & \frac{\partial f_{n}}{\partial \beta_{3}} \end{array}\right].$$

Equation (14) can be obtained by differentiating parameters $\beta_{1},\beta_{2,},\beta_{3}$:(16)$$\frac{\partial F}{\partial \vec{\beta }}=\sum_{p=1}^{n}2f_{p}(\vec{\beta })\frac{\partial f_{p}}{\partial \vec{\beta }}=2\left[\begin{array}{cccc} \frac{\partial f_{1}}{\partial \beta_{1}} & \frac{\partial f_{2}}{\partial \beta_{1}} & \cdots & \frac{\partial f_{n}}{\partial \beta_{1}}\\ \frac{\partial f_{1}}{\partial \beta_{2}} & \frac{\partial f_{2}}{\partial \beta_{2}} & \cdots & \frac{\partial f_{n}}{\partial \beta_{2}}\\ \frac{\partial f_{1}}{\partial \beta_{3}} & \frac{\partial f_{2}}{\partial \beta_{3}} & \cdots & \frac{\partial f_{n}}{\partial \beta_{3}} \end{array}\right]\left[\begin{array}{c} f_{1}(\vec{\beta })\\ f_{2}(\vec{\beta })\\ \vdots \\ f_{n}(\vec{\beta }) \end{array}\right].$$

Thus, the gradient of *F* (denoted $g_{F}$) can be expressed as
(17)$$g_{F}=2J^{T}f.$$

Assuming $f_{p}(\vec{\beta })\in C^{2}$, continuing the second-order partial derivative of Equation (16) yields
(18)$$\frac{\partial^{2}F}{\partial \beta_{i}\partial \beta_{j}}=2\sum_{p=1}^{m}\frac{\partial f_{p}}{\partial \beta_{i}}\frac{\partial f_{p}}{\partial \beta_{j}}+2\sum_{p=1}^{m}f_{p}(\vec{\beta })\frac{\partial^{2}f_{p}}{\partial \beta_{i}\partial \beta_{j}}.$$
where i,j=1,2,3. If the second term in the above equation is ignored, the following equation is obtained:(19)$$\frac{\partial^{2}F}{\partial \beta_{i}\partial \beta_{j}}\approx 2\sum_{p=1}^{m}\frac{\partial f_{p}}{\partial \beta_{i}}\frac{\partial f_{p}}{\partial \beta_{j}}.$$

Thus, the Hessian matrix of *F* (denoted $H_{F}$) can be expressed as
(20)$$H_{F}\approx 2J^{T}J.$$

The gradient of *F* and the Hessian matrix are now known, so the recurrence relation equation for the Gauss-Newton algorithm iteration is
(21)$${\vec{\beta }}_{k+1}={\vec{\beta }}_{k}-{\left(2J^{T}J\right)}^{-1}(2J^{T}f)={\vec{\beta }}_{k}-{\left(J^{T}J\right)}^{-1}(J^{T}f),$$
where ${\vec{\beta }}_{k}$ is the result of the previous iteration and ${\vec{\beta }}_{k+1}$ is the latest result obtained. When a specific condition for the iterative convergence of the Gauss-Newton algorithm is satisfied (e.g., $\left|F_{k+1}-F_{k}\right|$ is less than $10^{-6}$), the result is ${\vec{\beta }}_{k}^{*}$.

### 2.4. Matthews and Davies Algorithm

In the previous section, if $H_{F}$ is a singular value matrix, then the Gauss-Newton algorithm fails. Therefore, in this paper, we use the Matthews and Davies algorithm to correct $H_{F}$ and then apply the Gauss-Newton algorithm for APC. The Matthews and Davies algorithm is based on the Gaussian elimination method. This method leads to the modification of $H_{F}$. Additionally, it compares with other time-consuming algorithms, such as the Goldfeld, Quandt and Trotter algorithm and the Zwart algorithm, and is one of the most practical to use. For a Hessian matrix ($H_{F}$), its diagonal matrix D can be derived as
(22)$$D=LH_{F}L^{T},$$
where $L=E_{n-1}\cdots E_{2}E_{1}$ is a unit lower triangular matrix and $E_{1},E_{2},\cdots ,E_{n-1}$ is a primitive matrix. If $H_{F}$ is positive definite, then matrix D is also positive definite, and vice versa. If matrix D is not positive definite, the positive definite diagonal matrix $\hat{D}$ can be formed by replacing every zero or negative element in matrix D with a positive element, thus forming the positive definite matrix ${\hat{H}}_{F}$:(23)$${\hat{H}}_{F}=L^{-1}\hat{D}{\left(L^{T}\right)}^{-1}.$$

Let ${\hat{d}}_{k}=-{\hat{H}}_{F}g_{F}$ be obtained by parallelizing cubic Equation (23):(24)$$L^{-1}\hat{D}{\left(L^{T}\right)}^{-1}d_{k}=-g_{k}.$$

If we set $y_{k}=\hat{D}{\left(L^{T}\right)}^{-1}d_{k}$, it can be deduced from Equation (24) that
(25)$$L^{-1}y_{k}=-g_{k}.$$

In summary, the following can be obtained:(26)$$d_{k}=L^{T}{\hat{D}}^{-1}y_{k}.$$

Therefore, the recurrence relation equation for the final Gauss-Newton algorithm iteration after optimization is
(27)$${\vec{\beta }}_{k+1}={\vec{\beta }}_{k}+d_{k}={\vec{\beta }}_{k}-L_{k}^{T}{\hat{D}}_{k}^{-1}y_{k}.$$

### 2.5. Evaluation of Parameter Estimation

In parameter estimation problems, the variance of any unbiased estimator determines the CRLB, which is a lower bound, i.e., an unbiased estimator’s variance cannot be less than the CRLB and it can only ever approximate the CRLB indefinitely [31,32]. Here, as a performance indicator to serve as a guide, the CRLB of the MLE of the suggested complex exponential function is produced prior to examining the parameter estimate. The CRLB is an important criterion for evaluating parameter estimation [33] and is denoted the main diagonal element of the inverse of the Fisher information matrix (FIM) [34]. The following formulas can be used to determine the (m_th_, n_th_) element of the FIM (in terms of the proximal model):(28)[I(β→)]mn=tr[Cw−1(β→)∂Cw(β→)∂βmCw−1(β→)∂Cw(β→)∂βn]+2Re[∂s→H∂βmCw−1(β→)I∂s→∂βn].

There are three model parameters that need to be estimated, so the FIM has three orders. Since the noise is assumed to be Gaussian white noise with a zero mean and variance *σ*^2^, Equation (28) can be rewritten as
(29)[I(β→)]mn=2Re[∂s→H∂βmCw−1(β→)I∂s→∂βn]=2σ2Re[∂s→H(i)∂βm∂s→(i)∂βn].

Therefore, the FIM can be constructed according to Equation (29):(30)[I(β→)]11=2σ2Re[∑i=1N∂s→H(i)∂β1∂s→(i)∂β1]=2σ2∑i=1Nri2[I(β→)]22=2σ2Re[∑i=1N∂s→H(i)∂β2∂s→(i)∂β2]=2σ2∑i=1Nli2[I(β→)]33=2σ2Re[∑i=1N∂s→H(i)∂β3∂s→(i)∂β3]=2σ2∑i=1N(rihi)2[I(β→)]12=[I(β→)]21=2σ2Re[∑i=1N∂s→H(i)∂β1∂s→(i)∂β2]=2σ2∑i=1N(ri2θi)[I(β→)]13=[I(β→)]31=2σ2Re[∑i=1N∂s→H(i)∂β1∂s→(i)∂β3]=2σ2∑i=1N(ri2hi)[I(β→)]23=[I(β→)]32=2σ2Re[∑i=1N∂s→H(i)∂β2∂s→(i)∂β3]=2σ2∑i=1N(rihi)2.

Based on the matrix elements in Equation (30), the following expression for the FIM can be obtained:(31)$$\left[I(\vec{\beta })\right]=\frac{2}{\sigma^{2}}\left[\begin{array}{ccc} \sum_{i=1}^{N}r_{i}^{2} & \sum_{i=1}^{N}\left({r_{i}}^{2}\theta_{i}\right) & \sum_{i=1}^{N}\left({r_{i}}^{2}h_{i}\right)\\ \sum_{i=1}^{N}\left({r_{i}}^{2}\theta_{i}\right) & \sum_{i=1}^{N}{l_{i}}^{2} & \sum_{i=1}^{N}{\left(r_{i}h_{i}\right)}^{2}\\ \sum_{i=1}^{N}\left({r_{i}}^{2}h_{i}\right) & \sum_{i=1}^{N}{\left(r_{i}h_{i}\right)}^{2} & \sum_{i=1}^{N}{\left(r_{i}h_{i}\right)}^{2} \end{array}\right].$$

According to Equation (31), the FIM is determined by the variance *σ*^2^ of the Gaussian noise and the number of PSs, slant distance, azimuth and DEM. *σ*^2^ can be obtained from Equation (31) based on the *SNR*. Therefore, it is possible to obtain ${\left[I(\vec{\beta })\right]}^{-1}$, and the CRLB of the parameter to be estimated is the diagonal element of ${\left[I(\vec{\beta })\right]}^{-1}$.
(32)$$SNR=10\log_{10}\frac{0.5}{\sigma^{2}}.$$

The optimal values of the iterations, i.e., the parameter estimates, are obtained by setting the appropriate initial values of the iterations and the number of iterations, after which the *MSE* is introduced to evaluate the model error, and the *MSE* of the proximal correction model is computed as follows:(33)$$MSE=\frac{1}{N}\sum_{i=1}^{N}{\left[\Delta \phi_{i}-(\beta_{1}r_{i}+\beta_{2}l_{i}+\beta_{3}r_{i}\cdot h_{i})\right]}^{2}.$$

### 2.6. Algorithmic Process

The algorithm used in this paper’s method is shown in Algorithm 1.

**Algorithm 1:** Maximum likelihood Gauss–Newton iterative APC algorithm.**Input:** Δφi, ri, θi, hi**Output:**β→*=β→k+1, fp(k+1)(β→*) and Fk+1, (p=1,2,⋯,n)1:   Initialize k=0, β→k=β→0, ε and f(β→)=s→Hx→+x→Hs→=2cos(β1ri+β2li+β3ri⋅hi−Δφi)2:   **While** iteration does not converge **do**3:     Calculate fpk=fp(β→k) and Fk, (p=1,2,⋯,n) 4:     Calculate Jk, gFk=2JkTfk and HFk=2JkTJk5:     Calculate yk=−Lkgk and dk=LkTD^k−1yk6:     Calculate β→k+1=β→k+dk7:     Calculate fp(k+1) and Fk+1, (p=1,2,⋯,n)8:     **If** Fk+1−Fk<ε **then**9:       **break**10:      **Else**
k=k+1
11:      **end if**12:   **End while**

### 2.7. Experimental Information

In this experiment, a monitoring scene was selected in the Chair Hill area of Chongqing. To accurately capture the surface deformation within the region, the hillside was continuously monitored, and site photographs are shown in Figure 1. During the continuous monitoring process, we used the advanced slide radar GB-SAR technology. This radar system is capable of measuring small displacements of the ground surface by collecting radar signals without destroying the ground surface. It has a very high spatial and temporal resolution and can effectively identify small-scale signs of surface movement, which is important for the study of topographic microstructural changes and geologic hazards. The overall framework of APC involved in the method of this paper clearly demonstrates the flow of the whole correction process, as shown in Figure 2. This comprehensive method is expected to provide more accurate terrain deformation monitoring data and strong support for related scientific research.

## 3. Results

### 3.1. Adaptive Selection of PS Points

During this experiment, an advanced GB-SAR system was utilized to continuously monitor complex and variable hillsides. With 25 acquired radar images, Figure 3a demonstrates the polarimetric imaging effect presented in the area, from which the distribution of pixel points in the monitoring area can be clearly observed, and the amplitude of most of these pixel points is in the range of [−40, 0] dB. By setting the ADI threshold to 0.25, a total of 41,402 PS points were successfully identified (as shown in Figure 3c).

As seen from the above figure, there are more noise points in the PS selected by the amplitude deviation method, and there are cases in which the wrong PSs are selected, such as the pixel points in the red box in the figure. These noise points themselves have small amplitude deviation values, thus causing the incorrect selection of PSs. Therefore, to address the low *SNR* of the noise points, non-local mean filtering is used to improve the *SNR* of the SAR image, and then the *SNR* and coherence factor of the pixel points are calculated separately. Then, the K-means clustering algorithm adapts itself to calculate the thresholds to filter out the high-quality PSs while eliminating the noise points. The red box in Figure 3a–d indicates that there are obvious noise points in the area before processing.

Based on the classification criteria of the *SNR* and coherence factor, the results of the K-means clustering algorithm are shown in Figure 4, where the SC profile coefficient (ranging from 0 to 1) is 0.67785, indicating a good clustering effect. The three colors indicate that the clustering results are classified into three categories. Red, yellow and blue indicate noise points, low-threshold PSs and high-threshold PSs, respectively. An *SNR* threshold of −39 dB and a coherence factor threshold of 0.91 are adaptively calculated by the K-means clustering algorithm, and 20,197 high-quality PSs are selected by the threshold. The magnitude deviation index plot of the high-quality PSs is shown in Figure 5a, where the red box represents the area where the noise points have been significantly eliminated after processing, and Figure 3c shows that some of the noise points are eliminated. Figure 5b shows the differential interferogram of high-quality PSs, where the black circle indicates that there is a significant phase error in the region after differential interference.

### 3.2. Differential Interference

According to the selected high-quality PSs and the 1,414,930 point clouds from the TLS, the K nearest neighbor (KNN) algorithm is used for alignment, and the DEM values of the aligned point clouds are assigned to the PS. Afterward, Lagrange interpolation is used to calculate the DEM values of the remaining high-quality PSs. The results after alignment interpolation are shown in Figure 6. The K-means clustering algorithm is again utilized to divide the observation region into proximal and distal regions (shown in Figure 7). On this basis, multiple differential interferograms are obtained by differential interferometry using a 0 cm baseline with the previous image as the main image and the latter image as the auxiliary image. Figure 8 shows differential interferograms of the proximal and distal ends of the observation region.

### 3.3. Correction Results

Based on the method proposed in this paper, the maximum number of iterations is set to 1000, the initial value is ${\vec{\beta }}_{0}=(0,0,0)$, and APC is performed for the far and near observation regions. After Gauss-Newton iteration of the parameters, the approximate convergence values of the three parameters are ${\vec{\beta }}_{k}=(4.6\times 10^{-6},-1.7\times 10^{-5},-1.2\times 10^{-7})$. In the parameter estimation, the CRLB is a theoretical lower limit, while the MLE is a regularly used estimate technique. To verify the accuracy of parameter estimation in the case of a lower SNR, through 10,000 Monte Carlo simulation experiments and setting the *SNR* range to [−10, 6] dB, the *MSE* results of the MLE differ from those of the CRLB, and the estimation results are not accurate enough. Additionally, with the gradual increase in the *SNR*, the *MSE* results of the MLE gradually approach the CRLB (as shown in Figure 9), and the estimation results become more accurate. The differential interferometric phase and scatter plots of the overall observation area after correction are shown in Figure 10. The method in this paper can effectively correct for the APS relative to the traditional correction method. The reason for the poor correction effect of the traditional method may be that it is difficult for the general linear model to correct the atmospheric phase with null variability. At the same time, Figure 10a shows that the general PS point selection technique in the traditional method has difficulty removing noise points and thus affects the APC effect poorly.

To comprehensively assess the effectiveness of atmospheric phase correction, six representative reference points were arbitrarily selected in this study within the monitoring area; these reference points were distributed at different distance directions and azimuthal positions. Specifically, these six reference points were randomly assigned to the distal region, the proximal region, and the low-threshold region, with two points in each region, which were utilized for differential interferometric phase timing analysis.

By analyzing the differential interferometric phase timing diagrams, it is possible to visualize how the phase timing changes before and after correction. The exact locations of these reference points are marked in Figure 11, and the data after the APC process are shown. By comparing the post-correction time-series phases with those of the pre-correction and general correction methods based on the linear skew-distance model, the phases of all the reference points are improved, and the overall trend is closer to zero, which suggests the effectiveness of the phase correction method in this paper. By comparing Figure 12a,d, where the yellow color represents the statistical distribution histogram of PS phase and the red line represents the Gaussian fitting curve, the overall trend shows that the phase values corrected using this paper’s method tend to be closer to the true values, and the main value interval is distributed in the range of [−0.5, 0.5] rad, which further suggests the effectiveness of this paper’s phase correction method. Compared with the APC method based on the LSM and the Newton method (shown in Table 1), this paper’s method is superior.

## 4. Discussion

(1)Modeling assessment: By comparing the results of Figure 8 and Figure 10, we analyzed that the reason for the deterioration of the correction results using conventional methods may be that the conventional methods did not remove the influence of noise points, and the conventional method we used was a linear model that could not correct for the non-homogenous APS. Additionally, the influence of noise points resulted in the deterioration of the correction results. Therefore, a far-end and near-end nonlinear APC model is proposed. For the proximal observation scenario with lower terrain and a shorter observation distance, a proximal APC model based on a polar coordinate system plus the DEM in the vertical direction is designed in this paper. The distal atmospheric correction model is designed for the distal observation scenario with high terrain and long observation distances in the right-angle coordinate system, which considers the non-homogenous atmosphere in three directions. The validity of the model is verified by a test in the Chair Hill area of Chongqing. The results show that the method can improve the accuracy of GB-SAR atmospheric phase, thus greatly improving slope deformation estimation.(2)Error source analysis: A thorough examination of the algorithm employing the maximum likelihood approach in the presence of Gaussian white noise is a prerequisite for the theoretical derivation of the approach suggested in this work. Since the GB-SAR data are based on the PS technique, they suffer from extremely minimal noise interference and may be represented by a Gaussian distribution. The results suggest that the proposed strategy is reasonable and practicable. Furthermore, by observing Figure 12 and Table 1, it is evident that the correction effect of the Newton method is not satisfactory. We analyze that the reason may be the presence of matrix irreversibility in the Newton method, which leads to increases in correction error. Therefore, this also proves the superiority of the method proposed in this paper.(3)Optimization methodology: When there are multiple extremes, the derivative of the objective function will have multiple zeros, which facilitates the algorithm’s convergence to local extremes. The iterative algorithm’s ultimate goal is to find the parameter values corresponding to the maximum values. In the case of objective functions with multiple extreme values, the choice of initial values is very important. In particular, an accurate iterative solution is obtained when the iterative initial value is taken to approximate the global extremum, thus providing an accurate iterative solution for parameter estimation. However, how to accurately set the iterative initial value has been a difficult problem for the algorithm, so accurately setting the iterative initial value is a problem that needs to be solved in subsequent work.(4)The differences and similarities of research methods: The similarity between the research method in this paper and research method of Liu et al. is that both divide the experimental area into two parts based on the distance between the PS points and the radar center aperture [27], and they use different APC models to perform APC on the experimental area. The difference is that this paper uses the MLE method and the Gauss-Newton method to iteratively estimate the parameters of the APC model, which is an innovation compared to the traditional LSM. In addition, the similarity between this paper’s method and Mo et al.’s research method [29] is that both use the MLE method and iterative parameter estimation methods. The difference is that the correction of repositioning errors by Mo et al. is mainly related to azimuth, but this paper mainly studies APC, which is therefore related to slant range.

## 5. Conclusions

In this paper, a novel methodology for estimating parameters of the APC model based on MLE and the Gauss-Newton algorithm is proposed, and the superiority and practicability of the method in GB-SAR deformation monitoring are verified via simulation experiments and measured data. First, the amplitude value, amplitude deviation index value and coherence factor value of the pixel points in the radar image are calculated, after which the problem of artificially setting the thresholds of the amplitude value and coherence factor value is solved by the K-means algorithm for the adaptive selection of PSs. The high-quality PSs are classified into proximal and distal PSs according to the distance from the monitoring target to the radar site by the K-means algorithm, and the far and near ends are proposed. The nonlinear APC model, the near-end APC model based on the polar coordinate system, and the vertical direction of the DEM handle the non-homogenous APS at near distances, and the far-end APC model based on the right-angle coordinate system handles the non-homogenous APS at far distances in three directions (one vertical direction and two horizontal directions). Second, the objective function of the far-end and near-end correction parameter model is derived based on the MLE; the parameters of the objective function are iteratively estimated by the Gauss-Newton algorithm with suitable initial values, and the Gauss-Newton algorithm is optimized by the Matthews and Davies algorithm to improve the accuracy of parameter estimation. The *SNR* is in the range of [−10,6] dB through Monte Carlo simulation experiments when the *MSE* of the parameters gradually approaches the CRLB, which suggests the accuracy of the method in this paper for parameter estimation. Finally, the method is compared with the traditional APC method and Newton method based on real terrain data, which suggests the effectiveness and superiority of the method.

The theoretical innovation of this paper lies in combining the parameter model to be estimated in APC with MLE to establish a novel parameter model to estimate the likelihood function of APC, using the Gauss-Newton algorithm for the optimal parameter estimation and evaluating the accuracy of parameter estimation through Monte Carlo simulation experiments. This method avoids the problem of phase unwrapping and overcomes the limitations of the traditional LSM. Finally, this method is applied to monitor mountainous terrain deformation.

## Figures and Tables

**Figure 1 sensors-24-05699-f001:**
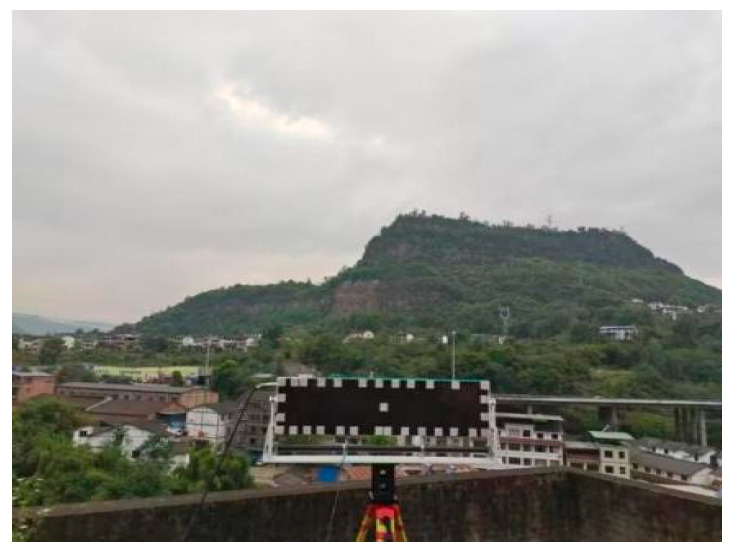
Scene diagram.

**Figure 2 sensors-24-05699-f002:**
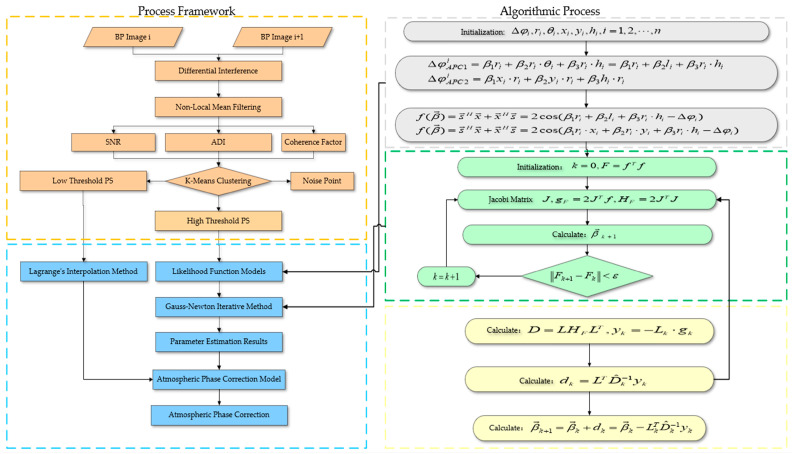
Overall framework of the methodology of this paper.

**Figure 3 sensors-24-05699-f003:**
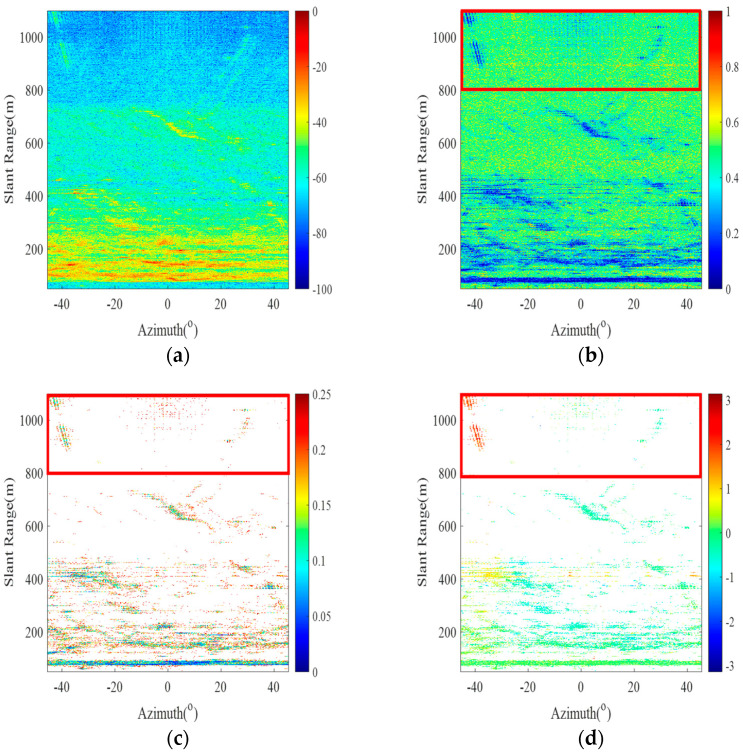
Radar data processing results. (**a**–**d**) The radar imaging map, amplitude divergence index map, PS amplitude divergence index map, and PS differential interferogram, respectively.

**Figure 4 sensors-24-05699-f004:**
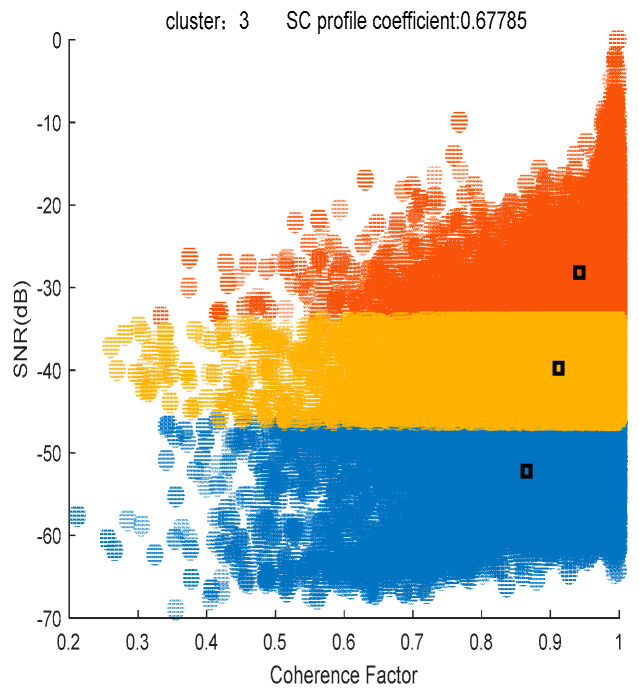
K-means clustering result plot.

**Figure 5 sensors-24-05699-f005:**
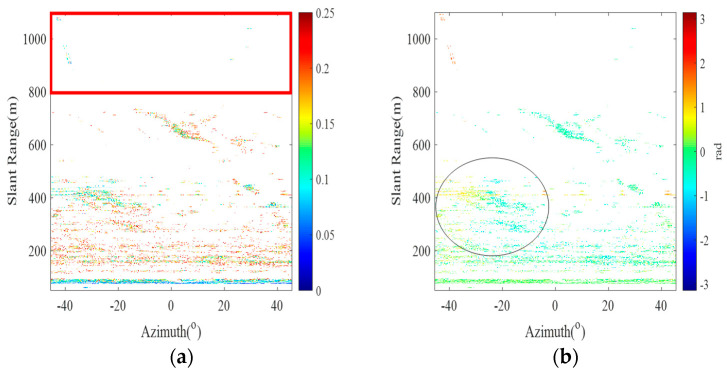
K-means clustering adaptive selection of high-quality PS results. (**a**,**b**) High-quality PS amplitude deviation index plots and high-quality PS difference interferograms, respectively.

**Figure 6 sensors-24-05699-f006:**
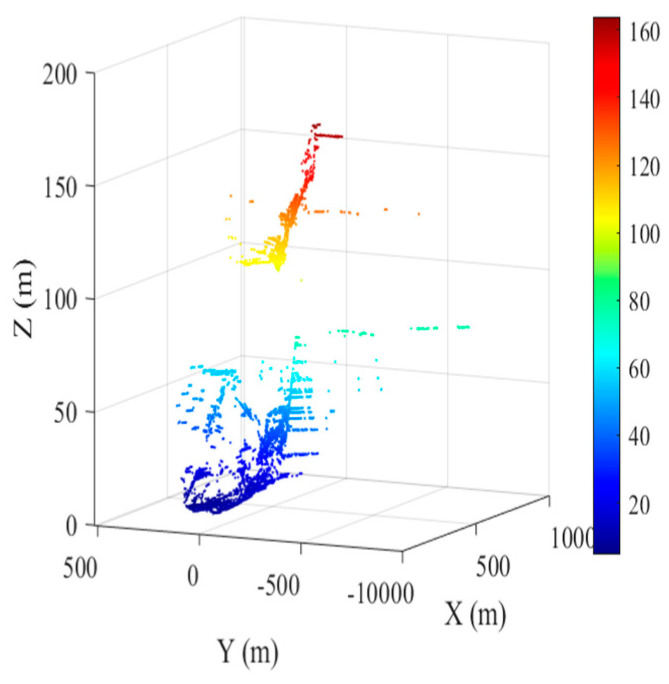
Alignment interpolation three-dimensional diagram.

**Figure 7 sensors-24-05699-f007:**
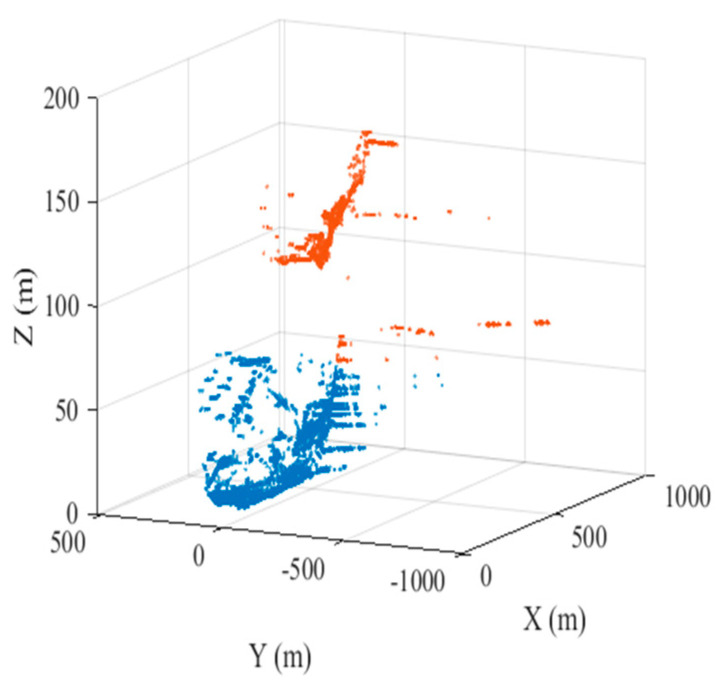
Three-dimensional scatter plot of distal and proximal clustering.

**Figure 8 sensors-24-05699-f008:**
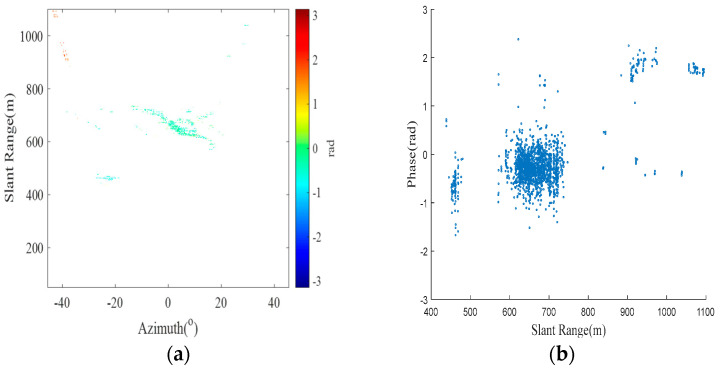
Differential interference result plots. (**a**,**b**) The differential interferometric phase and scatter plots at the far end of the 0 cm baseline, respectively. (**c**,**d**) The differential interference phase and scatter plots at the proximal end of the 0 cm baseline.

**Figure 9 sensors-24-05699-f009:**
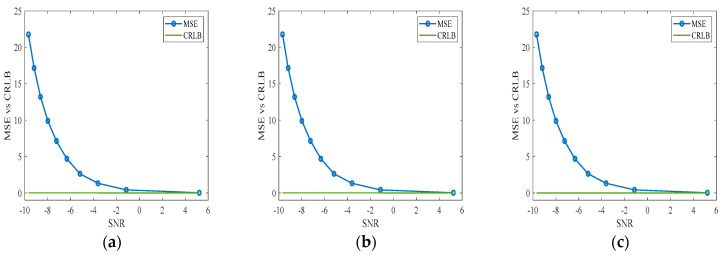
*SNR*-*MSE* plots. (**a**–**c**) The results of comparing the estimated *MSE* with CRLB for the β_1_, β_2_ and β_3_ parameters in the proximal correction model of the Monte Carlo experiment.

**Figure 10 sensors-24-05699-f010:**
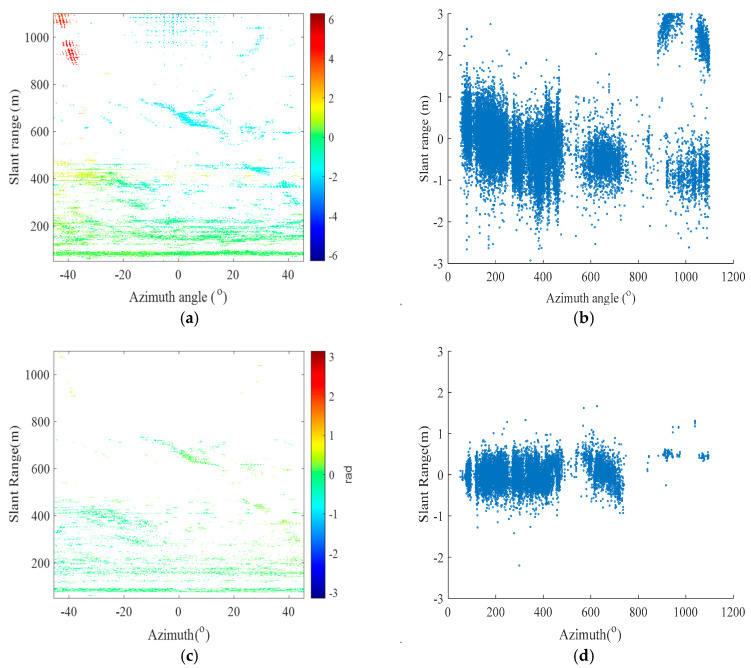
Results after correction. (**a**,**b**) The overall differential interferometric phase and scatter plots after correction by the conventional method, respectively. (**c**,**d**) The overall differential interference phase and scatter plots after correction at 0 cm baseline.

**Figure 11 sensors-24-05699-f011:**
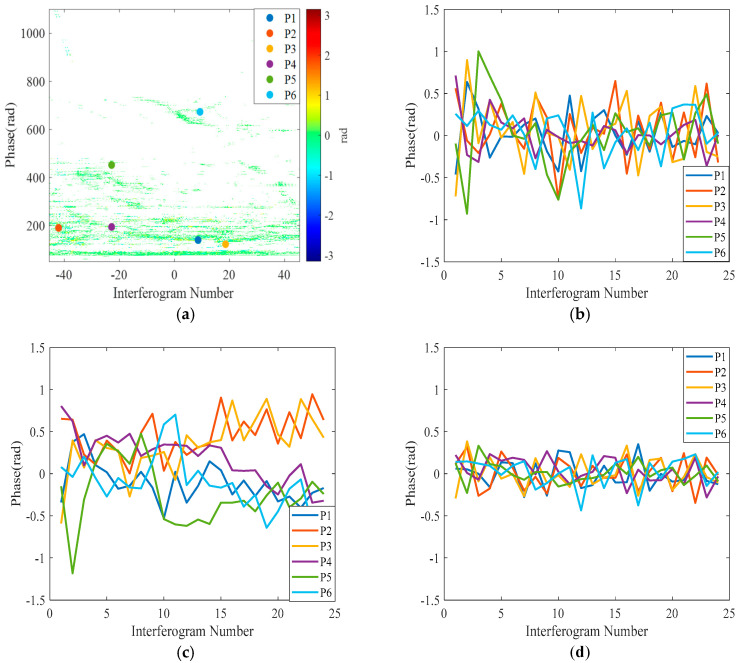
Differential interferometric phase timing diagram. The position of six reference points in the monitoring area are marked in (**a**–**d**). They are the differential interferometric phase timing diagrams for pre-correction, the conventional method, and the method of this paper, respectively.

**Figure 12 sensors-24-05699-f012:**
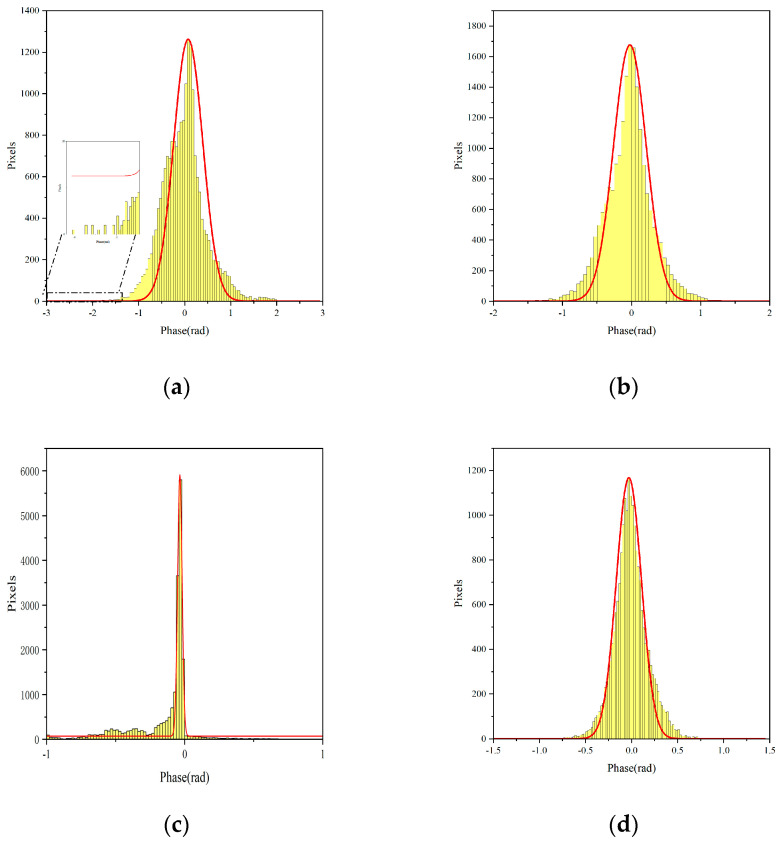
Histogram of frequency distribution. (**a**–**d**) The histograms of frequency distributions for pre-correction, LSM, the Newton method and this paper’s methods, respectively.

**Table 1 sensors-24-05699-t001:** Comparison of the results after correction by different methods.

Evaluation Indicators	Pre-Correction (Rad)	LSM (Rad)	Newton Method (Rad)	Methodology of This Paper (Rad)
Mean Value	0.1927	0.0458	0.1320	0.0092
Standard Deviation	0.4881	0.3959	0.2177	0.1908
Medium Error	0.4906	0.4564	0.2253	0.1917

## Data Availability

The data are not publicly available due to privacy. The data in this study will also be used for other studies.

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
