# Peer review of "A Novel Methodology for GB-SAR Estimating Parameters of the Atmospheric Phase Correction Model Based on Maximum Likelihood Estimation and the Gauss-Newton Algorithm"

_sensors, 2024, doi:10.3390/s24175699_

Round 1

Reviewer 1 Report

Comments and Suggestions for Authors

Regarding this manuscript, I have a few questions that the author needs to address:

1. The author claims to propose a novel atmospheric phase correction (APC) method for the ground-based synthetic aperture radar data. However, in section 2.4, the author mentions that the Gauss-Newton algorithm will fail when the Hessian matrix is a singular value matrix. This manuscript proposes using the Matthews & Davies algorithm to correct the Hessian matrix and then applying the Gauss-Newton algorithm for correction.

Therefore, the focus of this manuscript is to improve the input parameters of the Gauss-Newton. I think the author needs to modify the title and abstract to better align with the content of this manuscript

2. In the section 2, where does the method used by the author come from, and why is this method used? I think it would be more convincing to add appropriate supplementary content and references here.

3. In the section 3.3, the author needs more comprehensive materials to support the advantages of your proposed method compared to previous methods. I think it would be better to further expand this part of the content.

4. The discussion section is based solely on the author’s own description and lacks relevant supporting evidence.

5. The annotation of the formula needs to be clear. For example, what is β in formal (3) and how can it be obtained? Furthermore, what is the purpose of writing ri.θi as li? According to the content of this manuscript, this β is the most critical parameter and needs to be explained clearly.

6. The abbreviations of professional terms in the manuscript need to be modified. Please provide the full names of abbreviations that first appear in the main text.

7. Various annotations in the figures need to be explained in the caption when they first appear, such as the red box in Figure 3, the color in Figure 4, the black circle in Figure 5, and so on.

Reviewer 2 Report

Comments and Suggestions for Authors

The authors have created a new method for atmospheric phase correction for GB-SAR datasets. The approach relies on the Gauss-Newton algorithm and maximum likelihood estimation. The authors have also used the Matthews & Davies method which optimizes the Gauss-Newton technique for greater data processing accuracy. Overall, the paper is well-written, and results are acceptable. However, there are too many equations and theories which need to be reduced. The authors should address the following comments:

Comments:

1.     The authors are requested to go through the manuscript once again to remove spacing and other indentation errors. For example, second line of the abstract, there must be a space before the bracket; line 20, “far-“; line 223: “i.e.,An”. The authors should remove all such errors in the manuscript.

2.     Abstract line 25, why does the word “superiority” have a larger font size?

3.     Introduction First line, there is no need to explain the acronym GB-SAR every time once the full form is mentioned before in the abstract.

4.     Introduction line 75: “Partition correction… results” too long sentence. Consider splitting it into two.

5.     Introduction line 85: “Additionally, researchers… non-homogenous APS” is another sentence that does not have a proper meaning and is too long. The authors should work on properly conveying the rationale so that the scientific community can understand it easily.

6.     In Line 150, the authors have written “bounded by w in the distance direction” providing more information about w

7.     The authors should provide more information on the Gaussian noise in the real and imaginary parts and how w1,I  and w2,I both take similar value?

8.     The authors have written 30 equations in the manuscript which is highly irregular. The theories and equations that are already established in literature does not add any credit to the authors. Provide only the equations that have been modified by the authors to some extent or any new equation. This is not a book to read theory, it is a research paper.

9.     Why the amplitude departure index threshold is set to 0.25? Provide studies that have used the same value successfully or elaborate.

10.  Label the X and Y axis in Figure 4.

11.  If a terrestrial laser scanner is used, it should be mentioned in the methodology.

12.  Comparing Figures 8 and 10, the method was good in correcting differential interferometric cases (compare a and b), but not so successful in differential interference (compare c and d). Any reason why this happened? How could it be improved? Provide some explanation in the discussion section about it. 

Reviewer 3 Report

Comments and Suggestions for Authors

A review for “A Novel Methodology for GB-SAR Correcting Atmospheric Phase Error Based on Maximum Likelihood Estimation and the Gauss-Newton Algorithm”

Ground-based synthetic aperture radar (GB-SAR) is an active microwave detection method derived from interferometric synthetic aperture radar (InSAR). It is distinguished by its small size, high flexibility, and high spatial and temporal resolution. This method is mainly used in monitoring the movement of glaciers, landslides, and subsidence of mines and civil structures. It is important to choose appropriate correction methods to increase the accuracy of deformation monitoring in GB-SAR under different environmental conditions. However, due to atmospheric phase error (APS), deformation monitoring accuracy in GB-SAR is reduced. To solve this problem, the problem of atmospheric phase error (APS) minimization arises.

Atmospheric phase error is one of the factors that reduce the accuracy of Ground-based synthetic aperture radar (GB-SAR). Atmospheric phase error (APS) is mainly caused by the complexation of the atmospheric phase screen due to sudden changes in atmospheric conditions.  In traditional methods, atmospheric phase correction (APC) is performed by unwrapping the first phase and then removing APS based on the least squares method (LSM). However, the common phase unwrapping method may result in an unwrapping error. In such cases, LSM is a difficult process to directly apply, especially because the permanent scatterers (PS) phase is involved.  Therefore, this article proposes a new methodology aimed at solving such a problem.

The purpose and content of the study correspond to the special issue of Radar Remote Sensing and Applications—2nd Edition. Studies that have attempted to address comparable issues have been examined, and details about them have been provided. The issues identified, the benefits and drawbacks of the approaches, and the proposed solutions have all been examined in the research under review.
The paper discusses the problem of improving GB-SAR deformation monitoring measurement accuracy as a result of the research. The paper discusses the problem of improving GB-SAR deformation monitoring measurement accuracy as a result of the research. A new approach based on the Gauss-Newton algorithm and maximum likelihood estimation (MLE) method is put forth to overcome the issue. That is, the Atmospheric Phase Screen (APS) was assessed and adjusted to increase the measurement accuracy of GB-SAR deformation monitoring. The methodology was implemented as follows:

The first stage. According to the distance between the radar location and the calculation monitoring object, high-quality PS points are divided into proximal PS points and distal PS points based on the K-means clustering method. As a result, a proximal APC model and a distal APC model are proposed. A near APC model was generated based on a polar coordinate system and elevation (DEM), and the model was used to eliminate inhomogeneous APS in the near distance. This model is suitable for surveillance scenes with relatively flat terrain and small radar range slopes. The long APC model is based on a rectangular coordinate system and is used to eliminate inhomogeneous APS over long distances in one direction (one vertical and two horizontal). This model is mainly suitable for observation scenes with relatively steep terrains and large radar distance gradients.

The second stage. The implementation of the objective function of the model of near and far distance correction parameters is determined based on the maximum likelihood estimation method. The parameters of the objective function were evaluated iteratively using the Gauss-Newton algorithm based on appropriate initial values. The Gauss-Newton algorithm is optimized by the Matthews & Davies algorithm. Cramer‒Rao lower bound and mean squared error (MSE) were used to evaluate the results of parameter estimation. Monte Carlo simulation experiments were used to compare indicators.

The experiments were carried out based on real relief data. Comparisons are made with the existing general APC method and the APC method based on the LSM methods. The results are visualized.

The literature presented in the article is formed by the purpose of the article.

The following comments and shortcomings may be noted during the review process.

1.      Expression on line 167 is being rewritten. No need to rewrite it. It should be deleted. It is given in the formula on line 164.

2.        The expression N () in lines 170, and 171 is not clearly shown. Which are real and imaginary parts of Gaussian noise. It is necessary to give a clear expression.

3.             When writing formulas, if there is another formula in the next line, you should put a (comma) in it. It is necessary to consider everything according to the rules of writing formulas.

4.             In lines 151, 165, 176, 201, 210, and similar, it is necessary to write without leaving a paragraph. The article is written like this.

5.            5. 2.5 Evaluation of the parameter estimation performance part is not written with alignment on both sides (Justify text).

6.            Literature references are written with words. For example, lines 34, 35, 36 can be cited. References in other lines are also written in this way. Everything should be reconsidered.

7.             At the end of formulas .(point) or , commas are not placed at all. It is not appropriate to write the text mathematically or stylistically. All formulas should be punctuated depending on whether they end or continue.

8.             Clustering was performed in the division into Proximal PS points and Distal PS points. The K-means method was used as a clustering method. Using other clustering methods, such as Hierarchical Clustering, DBSCAN (Density-Based Spatial Clustering of Applications with Noise), Gaussian Mixture Models (GMM), Agglomerative Clustering, Spectral Clustering, Mean Shift Clustering, etc., the best clustering method is selected and the result can be increased.

9.            9. 2.6. algorithmic process is written in lowercase. It should also be written academically.

10.       10. The algorithm presented in lines 248-264 should be written in a certain standard form. It is appropriate if MDPI articles use the listed forms. Many articles present the algorithm in pseudo-code form.

11.        K nearest neighbor (KNN) classifier was used for alignment in lines 310-311. However, there is no information about the training options for K-nearest neighbor (KNN). Because this algorithm is based on learning.

12.       The article should be re-edited according to the journal standard.

13.        Convergence of the Gauss-Newton method is not guaranteed in all instances. Therefore, it is necessary to show the evaluation of the approximation. If the condition is satisfied, then the method is effective.

Comments on the Quality of English Language

Carefully check all grammatical errors. Still, the English language should be improved. I suggest asking for help from a native English.

Round 2

Reviewer 1 Report

Comments and Suggestions for Authors

Thank you to the author for answering my questions. However, I found that there are still some issues in this manuscript that need to be modified.

1. Line 73-75. In addition, many atmospheric phase correction (APC) methods based on homogeneous atmospheric media are not valid due to the complex terrain and various environmental factors, so there are methods to correct the atmospheric phase. I did not correctly understand the meaning expressed by the author in this sentence. I think it would be better to make some modifications to ensure consistency with the context.

2. Line 96. Please provide the full name of D-InSAR.

3. What is the subscript A in formula (a)? Please explain in the text.

4. What are j, w, and n in formula (2)? Please explain in the text.

5. Lines 221-222. “Therefore, in this paper, we use the Matthews & Davies algorithm to correct HF and then apply the Gauss-Newton algorithm for APC.” Please explain why this method should be used instead of other methods.

6. Figure 3. What is the red box? It was already mentioned in the first review. Please explain in the text or the caption.

7. Figure 4. What are the classification criteria or basis for the three colors? Please explain in the text.

8. Figure 5. What are the red box and black circle? Please provide explanations in the text.

9. In section 4. Since it is a discussion, it is necessary to analyze the similarities and differences between this study and other studies. I think there is a lack of necessary references here.
